# Insights into the Molecular Mechanisms of Eg5 Inhibition by (+)-Morelloflavone

**DOI:** 10.3390/ph12020058

**Published:** 2019-04-16

**Authors:** Tomisin Happy Ogunwa, Emiliano Laudadio, Roberta Galeazzi, Takayuki Miyanishi

**Affiliations:** 1Department of Environmental Studies, Graduate School of Fisheries and Environmental Sciences, Nagasaki University, 1-14 Bunkyo-machi, Nagasaki 852-8521, Japan; 2Department of Life and Environmental Sciences, Università Politecnica delle Marche, 60131 Ancona, Italy; e.laudadio@staff.univpm.it (E.L.); r.galeazzi@staff.univpm.it (R.G.)

**Keywords:** (+)-morelloflavone, kinesin Eg5, molecular dynamics, inhibitors, molecular interaction, biflavonoid

## Abstract

(+)-Morelloflavone (MF) is an antitumor biflavonoid that is found in the *Garcinia* species. Recently, we reported MF as a novel inhibitor of ATPase and microtubules-gliding activities of the kinesin spindle protein (Eg5) in vitro. Herein, we provide dynamical insights into the inhibitory mechanisms of MF against Eg5, which involves binding of the inhibitor to the loop5/α2/α3 allosteric pocket. Molecular dynamics simulations were carried out for 100 ns on eight complexes: Eg5-Adenosine diphosphate (Eg5-ADP), Eg5-ADP-***S***-trityl-l-cysteine (Eg5-ADP-STLC), Eg5-ADP-ispinesib, Eg5-ADP-MF, Eg5-Adenosine triphosphate (Eg5-ATP), Eg5-ATP-STLC, Eg5-ATP-ispinesib, and Eg5-ATP-MF complexes. Structural and energetic analyses were done using Umbrella sampling, Molecular Mechanics Poisson–Boltzmann Surface Area (MM/PBSA) method, GROMACS analysis toolkit, and virtual molecular dynamics (VMD) utilities. The results were compared with those of the known Eg5 inhibitors; ispinesib, and STLC. Our data strongly support a stable Eg5-MF complex, with significantly low binding energy and reduced flexibility of Eg5 in some regions, including loop5 and switch I. Furthermore, the loop5 Trp127 was trapped in a downward position to keep the allosteric pocket of Eg5 in the so-called “closed conformation”, comparable to observations for STLC. Altered structural conformations were also visible within various regions of Eg5, including switch I, switch II, α2/α3 helices, and the tubulin-binding region, indicating that MF might induce modifications in the Eg5 structure to compromise its ATP/ADP binding and conversion process as well as its interaction with microtubules. The described mechanisms are crucial for understanding Eg5 inhibition by MF.

## 1. Introduction

Kinesins are among the most explored protein families in protein–ligand interaction studies, probably owing to their important roles in essential cellular processes, such as cargo transport, neuronal development, and cell division. The mitotic kinesins play particular functions in bipolar spindle assembly, centrosomes separation, and faithful chromosomal segregation into daughter cells during mitosis, and other roles in microtubule polymer dynamics as well as in signal transduction [1,2,3,4,5,6]. Mitosis is a highly coordinated biological process that has become very attractive for therapeutic intervention in chemotherapy [7,8,9]. The mitotic kinesin Eg5 is one of the targets in such interventions because of its interaction with the spindle during mitosis to form and stabilize the bipolar spindle architecture [10,11]. Inhibitors of Eg5 disrupt cell division processes, thereby interfering with cell proliferation, without any significant effect on other cytoskeletal processes [12,13,14]. This characteristic approves Eg5 as a potentially effective, safe, and selective target in cancer treatment. To date, a number of antimitotic drugs targeting Eg5 have been developed and tested in clinical trials [14,15,16]. Unfortunately, the results of these trials indicate the need for further study to discover more candidates with the capacity not only to interact with, but also to inhibit the enzymatic activity of Eg5 towards blocking tumor cell proliferation [17].

Previous studies have revealed binding pockets on Eg5 where inhibitors can interact. These binding locations include the ATP-binding site (active site). Only few compounds, such as thiazoles [18], are known ATP-competitive inhibitors that bind at the nucleotide-binding pocket of Eg5. Other recent ATP-competitive inhibitors, including the biphenyl derivatives [19], have been reported to associate with the enzyme between the α4 and α6 helices [20,21]. ATP-competitive inhibitors are less sought after as inhibitors of Eg5 because they tend to alter the function of other ATP-dependent proteins. Inhibitors that bind at the allosteric site, i.e., the hydrophobic loop5/α2/α3 pocket, are more preferable owing to their affinity and specificity [22,23,24]. The presence of an elongated loop5 [25]—a peculiar feature of Eg5—confers selectivity and suggests lower toxicity of inhibitors that bind the allosteric loop5/α2/α3 pocket in Eg5. Therefore, researchers have focused on this binding pocket with the aim to design or identify new Eg5 inhibitors. Interestingly, computational approach has proved suitable in predicting possible mechanisms of inhibition of new Eg5 inhibitors as well as depicting the binding mode and molecular interactions of such novel inhibitors [13,26,27].

Morelloflavone (MF), a biflavonoid isolated from *Garcinia* spp., has been reported for its antitumor effect in vivo [28]. Recently, our research team identified Eg5 as a possible target for the anti-proliferative effect of the biflavonoid [29]. MF inhibited Eg5 basal and microtubule-activated ATPase activities. The inhibitor also suppressed the microtubule gliding of Eg5 in vitro [29]. The inhibitory mechanism appears to involve direct binding to an allosteric site on Eg5 to structurally alter the ATP-binding pocket, thus inhibiting its enzymatic functions. In the current study, we investigated the dynamics of Eg5-MF interaction while the inhibitor occupies the loop5/α2/α3 binding pocket. We aimed at unraveling the mechanisms underlying the modulation of Eg5 function by the biflavonoid (Figure 1) using computational approaches as depicted in Figure 2.

## 2. Results and Discussion

Figure 1 shows the chemical structure of MF, a *Garcinia* biflavonoid having a luteolin–apigenin structure [30]. We investigated Eg5-MF complex stability, conformational and structural behaviors, and compared the results to those of known inhibitors of Eg5: ispinesib and ***S***-trityl-l-cysteine (STLC) [14,15]. In total, eight biosystems were prepared including (1) Adenosine diphosphate (ADP)-bound Eg5, (2) Adenosine triphosphate (ATP)-bound Eg5, (3) Eg5 in complex with ADP and STLC, (4) Eg5 in complex with ADP and ispinesib, (5) Eg5 in complex with ATP and STLC, (6) Eg5 in complex with ATP and ispinesib, (7) Eg5 in complex with ADP and MF and (8) Eg5 in complex with ATP and MF (Figure 1). Among these, the first two biosystems served as negative controls, the next four biosystems were used to sample known inhibitors of Eg5 for comparison to our compound, and the last two biosystems were the test complexes to investigate the activity of MF towards Eg5. From 100 ns molecular dynamics (MD) runs carried out on each of the biosystems, data were critically analyzed to gain insights into the effect of MF on the Eg5 structure and catalytic function.

### 2.1. MF Exhibits Stability in Complex with Eg5 at the Allosteric Pocket

First, we monitored the stability of all the complexes during the simulation timeframe, using the root means square deviation (RMSD) to observe when the trajectories plateaued and converged towards an equilibrium state. As shown in Figure 3, all the complexes stabilized around 50 ns and remained stable throughout the 100 ns simulation. Before reaching stability, the Eg5-ADP-MF complex showed fluctuation up to 1.03 ± 0.12 Å (Figure 3A). 

The RMSD value of Eg5-ATP-MF was relatively low, as the system converged around 1.01 ± 0.35 Å (Figure 3B). In the first 25 ns of the simulation trajectory, Eg5-ATP-MF had an RSMD comparable to that of Eg5-ATP. Thereafter, the RMSD increased to a level similar to that of Eg5-ATP-ispinesib and Eg5-ATP-STLC, and was maintained until the end of the simulation. We noted that the Eg5-ATP showed a very stable complex with the lowest RMSD (0.75 ± 0.05 Å) and converged within the shortest time (20 ns) (Figure 3B). The ADP-bound Eg5 was less stable than ATP-bound Eg5 as the biosystem converged after 20 ns (Figure 3A). The difference in the nucleotides structure, i.e., the presence of a γ-phosphate in ATP which is missing in ADP, may contribute to the variation in the RMSD values. The relatively high RMSD values of Eg5-ATP-inhibitor complexes compared to the ATP-Eg5 complex suggest a structural deviation and conformational modifications in Eg5 occasioned by the presence of the inhibitors (MF, STLC, and ispinesib). Among the ADP-bound complexes, the highest RMSD value was observed with MF (1.03 ± 0.12 Å). Early into the simulation, STLC- and ispinesib-bound Eg5 in the presence of ADP displayed comparable structural deviation, with a slight deviation at the end of the 100 ns timeframe (Figure 3A). Eventually, the RMSD of Eg5-ADP-ispinesb and Eg5-ADP complexes became converged at around 0.9 ± 0.08 Å and 0.88 ± 0.12 Å, respectively, whereas that of Eg5-ADP-STLC was reduced to approximately 0.85 ± 0.05 Å. On the other hand, variation was observed in the ATP-bound counterpart of the complexes within the first 25 ns (Figure 3B). Usually, Eg5 hydrolyses ATP to ADP and Pi at the nucleotide-binding site to facilitate the generation of a favorable configuration that can interact with microtubules [31,32]. Convergence of Eg5 complexes at 50 ns in the current study suggests the stability of the biosystems during the simulation [33].

To verify the stability of the complexes, we checked the biosystem structures in the presence and absence of inhibitors using radius of gyration (Rg) analysis. As presented in Figure 3C, the Rg of the Eg5-ADP complex without inhibitor initially rose from 2.05 nm to nearly 2.4 nm early into the simulation (20 ns), but then continuously decreased to approximately 2.26 nm at the end of the simulation. However, this value was apparently higher than that obtained for the inhibitor-bound Eg5-ADP complexes (Figure 3C), suggesting an inhibitor-induced compactness in Eg5. For instance, the Rg value for Eg5-ADP-MF was reduced from 2.14 nm to less than 2.07 nm at 60 ns and later slightly increased to 2.13 nm, which was the lowest among all complexes. In contrast, for the corresponding Eg5-ATP-MF complex, the Rg increased from 2.06 nm to 2.21 nm throughout the simulation (Figure 3D) which was lower than that of Eg5-ATP. The presence of γ-phosphate of ATP might play a role in the Rg increase. For STLC- and ispinesib-bound Eg5-ATP structures, the Rg values were also lower compared to Eg5-ATP. These results suggest that the inhibitor-bound Eg5-nucleotide structures are very compact. Focusing on the inhibitors; they were stably resident within the loop5/α2/α3 allosteric pocket (Figure 3E), and the integrity of the Eg5 structure was maintained during the MD simulation, despite the observed fluctuations and displacements [34].

### 2.2. MF Induces Eg5-Loop5/α2/α3 Pocket Closure in a Manner Comparable to STLC

STLC and ispinesib reportedly both bind the allosteric pocket formed by the loop5/α2/α3, despite having diverse chemical structures [23,35,36]. Based on an Eg5-MF model generated by molecular docking, it was predicted that MF also binds at the allosteric loop5/α2/α3 pocket, ~12 Å from the nucleotide-binding site [29]. Here, we sought to unveil the possible effect of MF binding on the loop5/α2/α3 pocket in comparison with the effects of STLC and ispinesib. We studied the opening or closing of the allosteric binding pocket in Eg5 by measuring the distance between selected residues located on α3 and loop5, which formed the binding pocket. First, we estimated the distance between Trp127 and Glu215—two important residues—at the loop5/α2/α3 pocket. Trp127 is one of the functionally essential residues on loop5 that has been implicated in a major way in the opening/closure of Eg5 allosteric pocket. Glu215, on the other hand, is located on α3 of the protein (Appendix A). As early as 20 ns into the simulation, the allosteric pocket of Eg5, in the absence of inhibitors (Eg5-ADP), began to open (as deduced from the eventual distance between residues Trp127 and Glu215) reaching 2.31 ± 0.2 Å, and failed to close until the end of the simulation (Figure 4A). Evidently, the presence of MF rapidly reversed the opening of the pocket; the Trp127 — Glu215 distance in the Eg5-ADP-MF complex decreased from 1.13 Å to as low as 0.45 Å (Figure 4A). Despite the apparent attempt by the binding pocket to return to its open conformation, the presence of the inhibitor successfully prevented the allosteric pocket from opening. Interestingly, the protein remained in this state until the end of the simulation. The STLC and ispinesib data revealed a relatively tight closure of the loop5/α2/α3 pocket, which may have implications for their potency. For Eg5-ATP complex, the loop5/α2/α3 pocket opened as early as 3.5 ns during the simulation, with an attempt to close at approximately 20 ns. However, this attempted closure was promptly reversed and the distance increased to 3.44 Å. All other attempts to return the binding pocket to its closed state, mainly at 18 ns, 22 ns, and 60 ns, resulted in large increase in the Trp127 — Glu215 distance (Figure 4B), indicating that the Eg5 allosteric pocket was open in the absence of inhibitors. However, binding of MF, STLC, or ispinesib induced the Eg5 allosteric pocket to switch from the open to the closed conformation.

This observation on the loop5/α2/α3 pocket of Eg5 was further substantiated by evaluating the distance between Tyr211 and Trp127. The distance between these residues decreased significantly in Eg5-ADP-MF and Eg5-ADP-STLC structures compared to Eg5 structures in the absence of inhibitors (Figure 4C), suggesting the trapping of loop5 in the downward position. Notably, these observations are compatible with various crystals of Eg5-inhibitor structures [23,37,38,39]. Furthermore, we estimated the RMSD of loop5 during the MD simulations in the presence and absence of inhibitors. The results revealed lower values for inhibitor-bound Eg5 structures, suggesting a ligand-induced tightening of the allosteric pocket (Figure 4D). The values were 0.73 ± 0.06 Å for Eg5-ADP, 0.62 ± 0.05 Å for Eg5-ADP-ispinesib, 0.4 ± 0.03 Å for Eg5-ADP-STLC, and 0.31 ± 0.02 Å for Eg5-ADP-MF. Finally, the Eg5-MF complex was deeply inspected, and the loop5 could indeed be seen in the downward position when MF was bound (Figure 4E). Analyzing the poses at the end of the MD simulations, we found a very similar reorientation for MF and STLC within the loop5/α2/α3 pocket. Both inhibitors moved closer to loop5 as confirmed from the interaction modes and the distance from loop5 (Figure 4F). This behavior marks the likely similar mode of action of these molecules, which is compatible with their similar binding configuration [29].

### 2.3. MF Induces Compactness and Stabilizes the Eg5 Allosteric Pocket

To better understand the local effect of MF and other inhibitors on the Eg5 loop5/α2/α3 pocket, we investigated the degree of movement associated with the binding site throughout the simulation in more detail, within a distance of 8 Å in all directions around the inhibitors. 

In this range, we found the residues Thr112, Phe113, Met115, Glu116, Gly117, Glu118, Arg119, Ser120, Trp127, Glu128, Leu132, Ala133, Gly134, Ile136, Pro137, Arg138, Leu171, Leu172, Tyr211, Ile213, Leu214, Glu215, Lys216, Ala218, Ala219, Lys220, Arg221, Thr222, and Phe239, which were used to prepare an index file. The RMSD plots for the allosteric pocket in the Eg5-ADP and Eg5-ATP complexes were unsteady, with high fluctuations of up to 0.6 Å (Figure 5A,B). This may be due to the presence of loop5 within the region. The elongated and conserved loop5 (residues Gly117–Gly133) significantly contributes to the creation of the hydrophobic cleft in conjunction with α2 and α3 helices to accommodate interacting inhibitors. Some authors have suggested that loop5 can regulate the rate of conformational change occurring at the nucleotide-binding pocket [33,39,40]. Notably, the presence of MF in the allosteric site decreased such fluctuations. The Eg5-ADP-STLC complex showed the lowest RMSD values and stabilized around 0.42 Å at 100 ns. The RMSD patterns obtained for Eg5-ATP-inhibitor complexes were similar to those of ADP-bound Eg5-inhibitor complexes. Next, we investigated the Rg of the allosteric binding site for all Eg5 complexes evaluated in this study (Figure 5C,D). We found that ispinesib, which showed the lowest impact on the RMSD of the putative allosteric pocket of Eg5, actually induced the highest degree of compactness on the binding cleft. The compactness was stronger at the allosteric pocket compared to the global protein structure in both ADP- and ATP-bound complexes (Figure 3C,D). We also observed that the compactness caused by MF binding to the loop5/α2/α3 binding pocket in the presence of nucleotides was stronger than that of STLC. Although the known inhibitors of Eg5 displayed varied structural properties, most of them still identified and bound the allosteric site to exert their inhibitory effects. Together, our data indicated that, in addition to inducing closure of the loop5/α2/α3 allosteric pocket, MF stabilizes and compacts the allosteric pocket as part of its interaction mechanisms.

### 2.4. Binding of MF on the Allosteric Site Alters the Eg5 Structural Conformation

To explore the structural behavior of Eg5 in the presence or absence of MF, we estimated the distance between selected regions, such as switch I, switch II, α-helix 4 and the central β-sheet (β6) of ADP-bound Eg5, and compared the data with those obtained for STLC and ispinesib. Switch I encompasses residues Met228–Ser235 (highlighted in yellow in Figure 5A), whereas switch II comprises Leu266–Asn289 (red in Figure 5B). β6 consists of His236–Met245 (orange in Figure 5B), and α-helix 4 encompasses Gln290–Val303 (light green in Figure 6A). For the analysis, an index file was created for each system, in which we recorded all the above residues, and monitored the distances between these regions throughout the MD simulation. As the switch I loop conformation is known to aid ATP binding through formation of a closed nucleotide-binding cleft [41], we evaluated the distance between switch I and α-helix 4 along the simulation trajectories. In the biosystem without inhibitor, the initial distance between these two domains was 0.58 Å. Twenty nanoseconds into the simulation, we observed an increase in this distance to 0.92 Å, which was retained throughout the rest of the simulation (Figure 6A). In the presence of inhibitors, switch I and α-helix 4 became very close to each other. In detail, we noted final distances of 0.54 Å, 0.22 Å, and 0.26 Å for the Eg5-ADP-ispinesib, Eg5-ADP-STLC, and Eg5-ADP-MF complexes, respectively. These results indicate that the inhibitors influence the motions of Eg5 structure, and that STLC and MF may have a similar inhibitory mechanism. When we analyzed the space between switch II and β6, we found a similar distance (size 0.25 Å) for all Eg5 complexes at the beginning of simulations (Figure 6B). After 25 ns of simulations, this distance reached 0.66 Å in Eg5-ADP-ispinesib and was maintained around 0.65 Å until the end of the simulation. This is a peculiar behavior, which suggests the likely different mode of action of ispinesib with respect to the other inhibitors studied. In fact, for the other complexes, final distances ranged between 0.23 Å and 0.27 Å.

Previous reports on wet experiments suggested that inhibitors that bind to the Eg5 allosteric pocket can induce an Eg5 conformation with weak affinity to microtubules [15,42,43,44]. The binding of MF to the allosteric pocket of Eg5 prompted us to evaluate the flexibility of ATP- and ADP-bound Eg5 complexes in the absence and presence of the inhibitor in order to understand how the compound might alter the overall protein conformation. The root mean square fluctuation (RMSF) profiles shown in Figure 6C,D showed a similar trend of residual fluctuations at different regions of Eg5 for all the complexes. The unsteady profile in the RMSF plots reflects the structural alterations in the highly flexible regions of the protein, which is consistent with its b-factor (Appendix A). For instance, the β-sheet that is buried in the core of Eg5 has lower RMSF values than solvent-exposed regions, such as the neck linker and the loop regions. Eg5-ATP displayed reduced flexibility at the N-terminus of the neck linker region when compared to Eg5-ADP (Figure 6C,D). 

In the Eg5-ADP structure, the switch I region showed higher fluctuation than in Eg5-ATP. This may be due to the absence of γ-phosphate in ADP, which is consistent with a previous study that revealed that the lack of γ-phosphate in ADP-bound Eg5 allows the switch I portion of Eg5 to move frequently during simulation [40]. Hence, the Eg5-ATP structure is more rigid than the Eg5-ADP complex. In this study, the fluctuations recorded for the switch II region were lower in Eg5-ATP than in Eg5-ADP. These observations are often associated with the repeated contacts formed by switch I with γ-phosphate of ATP while switch II afforded a β-hairpin conformation [33]. Since these interactions are missing in ADP-bound Eg5, it permits the Eg5-ADP structure to adopt a range of diverse conformations resulting in enhanced flexibility. Another probable explanation for the relatively rigid state of switch II in the Eg5-ATP complex is a hydrophilic interaction between γ-phosphate oxygen of ATP and the nitrogen moiety of Gly268 [40], which can position the switch II loop in a manner that might cause steric hindrance against Eg5-microtubule binding. Furthermore, a hydrogen bond between the oxygen moiety of the ATP ribose ring and the amide nitrogen of Asn29 possibly afforded switch II less flexibility [40]. It must be noted that this hydrogen bond is not formed in the presence of ADP since the missing γ-phosphate group in ADP allows the molecule to be placed slightly upward in the nucleotide-binding cleft. Hence, the ribose ring and amide nitrogen of Asn29 cannot interact [40]. It should be emphasized that switch I, switch II, and various other loops of Eg5, including loop5, loop7, and the p-loop, are functionally essential in the mitotic protein, and interference with their structural characteristics may affect the enzyme functions.

Relative to the inhibitor-free Eg5 structures, the presence of MF, STLC, and ispinesib distorted the flexibility of Eg5, indicating their potential to induce conformational change in the protein. The binding of MF to Eg5 was mostly associated with a reduction in fluctuation of Eg5. However, it appears that the tubulin-binding region gains flexibility of residues after MF interaction (Figure 6E), and MF may interfere with Eg5-microtubule interaction as one of its possible Eg5-inhibitory mechanisms. This is in agreement with our earlier report that MF significantly affects microtubule binding to Eg5 and inhibited its microtubule-activated ATPase activities [29].

### 2.5. Binding Free Energy Estimation for Eg5-Inhibitor Complexes

To quantify the effects of small modifications on the complexes, and provide a means to the partition of free energy, the ligand dissociations energies of Ispinesib, STLC, and MF in the Eg5 model were determined from umbrella sampling. Figure 7 shows that the ligand dissociation free energy barrier of ispinesib is significantly higher compared to those of STLC and MF. In fact, in the case of Eg5-ADP-ispinesib, the energy value is around 39 kcal/mol. MF shows a free energy barrier at 28.7 kcal/mol, higher than STLC (25.5 kcal/mol). This result suggests similar features between the Eg5-ADP-STLC and Eg5-ADP-MF complexes. Another interesting observation is the existence of a plateau region in the energy curve for all inhibitors. Analysis of structures from the simulations for ξ = 6 to 8 Å (for ispinesib) and ξ = 8 to 10 Å (for STLC and MF) shows that ispinesib presents a different temporary stabilized state relative to STLC and MF. The presence of these stabilized temporary phases for all three inhibitors is due to the different environment, and magnitude of the non-equilibrium contributions. Indeed, at a molecular level, every dissociation event in solution involves slightly different arrangements of atoms and therefore the kinetic data is not explained by a single barrier height but rather by a distribution of barriers. We can distingue three different states:Bound state (windows 0–5 for Ispinesib, 0–7 for MF, and 0–8 for STLC) where the inhibitors are tightly bound to Eg5 via their numerous charge contacts. This effectively restricts the inhibitors’ rotational freedoms such that similar behavior is observed across all trajectories.Transition state (windows 5–10 for Ispinesib, 7–10 for MF, and 8–10 for STLC) where inhibitors must undergo some rotation to break out. In this region, the interactions between inhibitors and Eg5 are gradually broken. This corresponds to increasing rotational freedom in the molecules.Electrostatic region (windows 10–14 for Ispinesib, 10–14 for MF, and 10–12 for STLC) where all direct contacts have been severed and residual electrostatic interactions have been screened by incoming water molecules. Under weak electrostatic attractions, the inhibitor molecules are more or less rotationally free.

Umbrella sampling results showed that MF has very similar energy values to STLC, a known Eg5 inhibitor. We also found that there are many correspondences between the barrier heights of Eg5-ADP-STLC and Eg5-ADP-MF, suggesting similar ways to obtain dissociation complexes. These data confirm that MF is a promising Eg5 inhibitor.

The Molecular Mechanics Poisson–Boltzmann Surface Area (MM/PBSA) method was also used to calculate the complexes overall free Gibbs energy (i.e., stability). Snapshots of trajectories from 100 ns simulations at every 10 ps of stable intervals were retrieved to serve as inputs for free energy determination. The binding free energies and their corresponding components for Eg5-inhibitor complexes are presented in Table 1. Generally, energy terms that contribute to protein–ligand complex formation are broadly categorized into polar (polar solvation and electrostatic) and non-polar (Van der Waals and non-polar solvation) energies [45]. In this study, electrostatic interactions, polar solvation energy, and van der Waals were negative whereas non-polar solvation was the only positive energy term for the complexes. This suggests that the major impairment to binding in all Eg5-inhibitor complexes was the non-polar solvation energy whereas electrostatic interactions, polar solvation energy, and van der Waals mainly facilitate ligand binding at the Eg5 allosteric site. In other words, Eg5-ADP-MF, Eg5-ADP-STLC, and Eg5-ADP-ispinesib complexes are mainly stabilized by electrostatic, van der Waals, and polar solvation energies.

Notably, electrostatic interaction was a predominant contributor to the total binding energy in all the complexes, whereas Van der Waals interaction energy contributed the least. For STLC and MF complexes, the contributions of all energy terms were comparable. This is consistent with data from umbrella sampling (Figure 7). However, the Van der Waals and polar solvation energies were much lower than those found for ispinesib. The binding energies of Eg5-ADP-Ispinesib, Eg5-ADP-STLC, and Eg5-ADP-MF were -96.31, -88.98, and -93.66 kJ/mol, respectively. These results indicate that Eg5 inhibitors bind tightly to the protein in a manner that may be sufficiently strong to induce structural modifications and interfere with the ATPase and motor activities of the enzyme. The energy values also indicated that the binding process was spontaneous at the loop5/α2/α3 pocket, lending credence to our prediction of the allosteric pocket as the actual binding site for MF. Among all the ligands investigated, ispinesib exhibited the highest affinity to the allosteric pocket of Eg5 (Table 1), revealing the favorable interaction between Eg5 and ispinesib, which may be implicated in its inhibitory potency (IC_50_ = 3 nM) [23].

### 2.6. MF Alters the Affinity of Nucleotides to the Active Site of Eg5

Figure 8A shows the predicted binding affinity of ADP on Eg5 in the absence and presence of inhibitors. It was previously shown that the presence of inhibitors changes the stability of Eg5-nucleotide complexes. Here, the binding energy of ADP was lowered by the inhibitors (Figure 8A). This may contribute to the suppression of ADP release from the Eg5 active site—an essential mechanism in Eg5 inhibition. The reduced affinity of Eg5 for nucleotides may weaken its affinity to microtubules. To better understand this behavior, we evaluated the stability of the active site by computing the RMSD and Rg for each biosystem in the absence and presence of the inhibitors.

The active site of Eg5 in complex with ATP had lower RMSD values and was relatively more stable than the ADP counterpart (Figure 8B). This is consistent with our previous finding of the reduced flexibility in the Eg5-ATP complex as occasioned by the presence of γ-phosphate. Lack of the γ-phosphate in ADP increases the number of potential conformations and thus, lowers the stability of the complex. STLC, ispinesib, or MF had varied effects on the stability of the nucleotide-binding site (Figure 8C–E). The Eg5-ATP-inhibitor complexes displayed greater stability for the active site than the Eg5-ADP-inhibitor as deduced from the RMSD plots. It is suggested that the binding of MF, STLC, or ispinesib to Eg5 drives the protein to adjust its allosteric pocket to accommodate the inhibitors. This consequent structural alteration can affect the stability of the nucleotide-binding site which is located roughly 12 Å away from the inhibitor-binding pocket. To obtain more clear information, we monitored the Rg of the ATP-binding pocket for the Eg5 complexes. The Rg values were higher than that of MF-bound Eg5 (Figure 8F), indicating that the inhibitor increases the compactness of the active site. A similar result was obtained for STLC, but not for ispinesib, which increased the Rg from 1.55 nm to 1.91 nm. This suggests that ispinesib decreases the degree of compactness in Eg5 at the nucleotide-binding pocket, highlighting again the probable different mode of action of ispinesib compared to MF.

## 3. Computational Methods

### 3.1. Starting Structures

The chemical structure of MF was built from data retrieved from the literature [30]. ChemAxon software [46] was used to prepare cleaned-up 2D-coordinates of the MF, which was converted to 3D geometry using the Conformers suit of the software based on the Merck molecular force field (MMFF94). The FASTA format of Eg5 (PDB ID: 3KEN) was retrieved from PubMed to model the starting protein structure on SwissModel server [47]. The 3D structure of modeled Eg5 was visualized with PyMOL [48] as a cartoon representation for observation of the co-crystallized ligands. All water molecules and ligands, except adenosine diphosphate (ADP), were deleted. Molecular docking was carried out using the modeled structure to obtain Eg5-ADP-MF complex as described previously [29]. To generate Eg5-ATP-MF complex, we simply replaced the co-crystallized ADP with ATP, with the adenine, ribose, and string of phosphate groups well aligned. Next, we deleted the bound MF molecule to obtain the control complexes (Eg5-ADP and Eg5-ATP). For comparison, crystal structures of Eg5 in complex with two potent inhibitors (STLC and ispinesib) were retrieved as Eg5-ADP-STLC (PDB ID: 3KEN) and Eg5-ADP-ispinesib (PDB ID: 4AP0), respectively, from the RCSB PDB Data Bank [49]. Missing segments of the protein structures were modeled. In addition, the cocrystallized ispinesib, which appeared to have lost the alkyl group, was substituted with a complete ispinesib structure obtained from the Macromodel MAESTRO suite. Finally, the cocrystallized ADP molecule in these structures was substituted with an ATP molecule to generate the Eg5-ATP-STLC and Eg5-ATP-ispinesib complexes. Autodock Vina on PyMOL was used for preparing these complexes [50,51].

### 3.2. Biosystems Setup

Energetic parameters for ADP and ATP were obtained from the AMBER parameter database [52]. Inhibitors parametrization was carried out using the general AMBER force field (GAFF) [53]. For the protein structures, the H^++^ server [54] was used to compute pK values of ionizable amino acids and the correct protonation state was assigned to the residues based on a pH value of 7. All the histidine residues were protonated at the epsilon position and all the aspartic and glutamic acids were retained in their anionic form. The protein and ligands were combined to obtain the biosystems for MD simulation starting from the docked cluster reported in our previous article [29]. A cubic periodic boundary condition was then set up corresponding to a cubic box 10 nm long in all directions and with the complex at its center. Solvation of all the biosystems in a transferable intramolecular potential three-point (TIP3P) water model [55] was carried out, followed by neutralization using Na^+^/Cl^-^ ions (0.15 M). Prior to proceeding to minimization and dynamics, the geometry of the ligands was optimized at the B3LYP/6-31G* level using G09 [56], to obtain their charges and the missing AMBER parameters [57,58]. The obtained system was then minimized with the steepest descent algorithm (10,000 steps) followed by 5,000 cycles using the conjugate gradient algorithm until the threshold (Fmax <100 kJ/mol) was reached.

### 3.3. MD Simulation

Groningen Machine for Chemical Simulations (GROMACS) version 5.0 [59,60] was used to run all atomistic simulation for trajectory analyses employing the AMBER-99SB-ILDN force field [61]. Equilibration was carried out using an accurate leap-frog integrator for equations of atomic motion with a time-step of 0.002 fs at constant number of particles, volume and temperature (NVT) for 200 ps and constant number of particles, pressure, and temperature (NPT) condition for 2 ns. The temperature was kept at 310 K using the V-rescale thermostat algorithm [62] with a short preliminary 200 ps run in the NVT ensemble, applying the positional restraints to the protein with a force constant of 1000 kJ/mol, for the whole NVT run while the Parrinello–Rahman barostat algorithm was used to maintain pressure at 1 bar [63]. During these steps, protein and ligand as well as water and ions were coupled to their own temperature and pressure while a full positional constraint was imposed on the heavy atoms in all directions using the Linear Constraint Solver (LINCS) algorithm for bond constraint [64]. The particle mesh Eldward (PME) algorithm was used to estimate the electrostatic interactions [65,66,67]. The cut-off range for electrostatic and Van der Waals interactions was set to 1.2 Å for both. Using the more accurate Nosè–Hoover thermostat and setting the time constant for coupling to 0.5 ps, production phase simulation was performed for 100 ns in NPT ensemble on each of the biosystems with removal of all positional restraints [68,69].

### 3.4. Umbrella Sampling and Data Analysis

MD trajectories generated during 100 ns were analyzed using GROMACS toolkit utilities. RMSF, RSMD, Rg and hydrogen bond distribution for each system were determined [70]. The *g_dist* tool was used to estimate the distance between residues [70]. Umbrella sampling simulations were used to determine the ∆G of binding [71] along the dissociation main axis that corresponds to the reaction coordinate ξ (rc) chosen as the distance between the center of mass (COM) of Eg5 inhibitor molecules and the midpoint of the line connecting the backbone atoms of amino acids that compose the inhibitors binding site. The ξ chosen has been proved to be suitable to describe the transition between the inhibitors-bound and the inhibitors-unbound structures. Umbrella sampling simulations were run between ξ = 16 Å using a harmonic force constant, k= 40 kcal/mol. For each sampling window, the 100 ns equilibrated Eg5-ADP-inhibitor complexes obtained after 100 ns were used for another 200 ps followed by another 1 ns MD for the umbrella sampling; the free energy profiles of the three studied systems were then compared and quantified. The changes in free energy along the reaction coordinate were calculated by using the weighted histogram analysis method (WHAM) [72]. Free binding energy (G_binding_) of Eg5-inhibitors complexes was determined by the MM/PBSA method using the g*_mmpbsa* tool [45]. Snapshots were extracted at every 10 ps to calculate the free Gibbs energy values [73,74,75]. PyMOL was used for visualization and interaction analysis of the docked complexes. Visual molecular dynamics (VMD) [76] and CHIMERA [77] softwares were employed for trajectory visualization and analyses, while Xmgrace (Grace 5.1.21) was used for generating the plots [78,79].

## 4. Conclusions

Eg5 is a validated target for antimitotic agents owing to its importance in cell division. As MF (an anticancer biflavonoid) was previously found to inhibit Eg5 activities, we investigated the dynamics of the Eg5-MF interaction at the molecular level using atomistic simulation to explain the inhibitory mechanism of the biflavonoid. MF displayed tight binding on the loop5/α2/α3 pocket of Eg5 in a manner similar to STLC and ispinesib, which implies that the allosteric pocket is its binding site on Eg5. MF binding induced closure of the allosteric pocket through trapping the loop5 in the “closed” conformation, comparable with the previously reported Eg5 crystal structures co-crystallized with potent inhibitors. Structural modifications of the tubulin-binding region in the presence of MF can compromise the normal binding mode of Eg5 with microtubules as one of the MF mechanisms for inhibiting Eg5 ATPase and motility functions. Upon MF binding, the Eg5 allosteric pocket becomes very compact and stabilized, suggesting a more rigid structure that can contribute to the trapping of Eg5 in the ADP-bound conformation.

Taken together, our data show that the inhibition of Eg5 activities by MF indeed involves a stable interaction with the protein at the putative allosteric pocket, and alteration of the conformation of the enzyme. In vivo experiments to confirm the inhibitory mechanisms of morelloflavone against Eg5 and crystallization of Eg5-nucleotide-MF will be undertaken in our subsequent research work to gain more insights on the evolution and biochemical implication of the molecular structure of the complexes.

## Figures and Tables

**Figure 1 pharmaceuticals-12-00058-f001:**
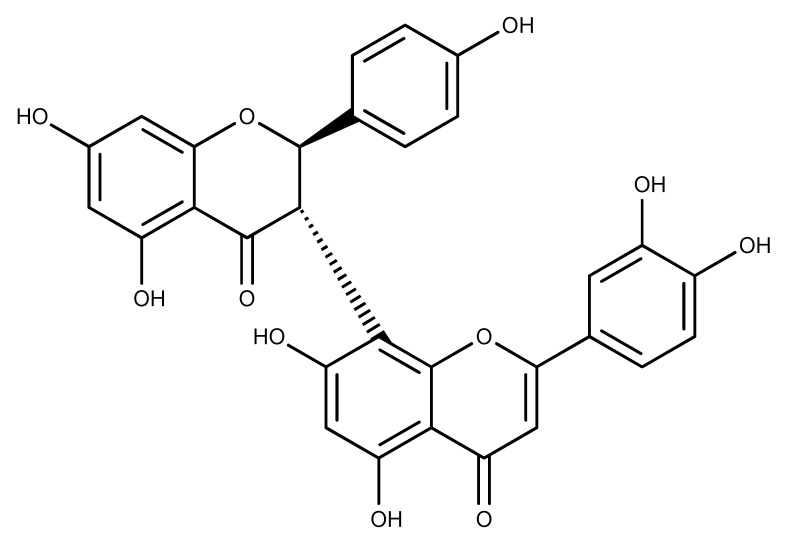
Chemical structure of (+)-morelloflavone [30].

**Figure 2 pharmaceuticals-12-00058-f002:**
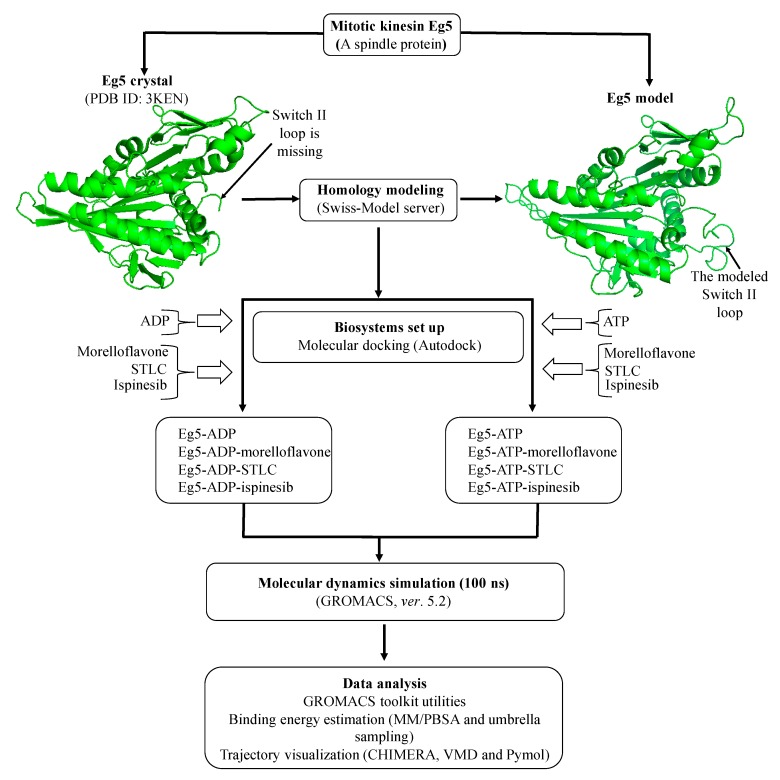
Flow-chart for the computational strategy used in this study.

**Figure 3 pharmaceuticals-12-00058-f003:**
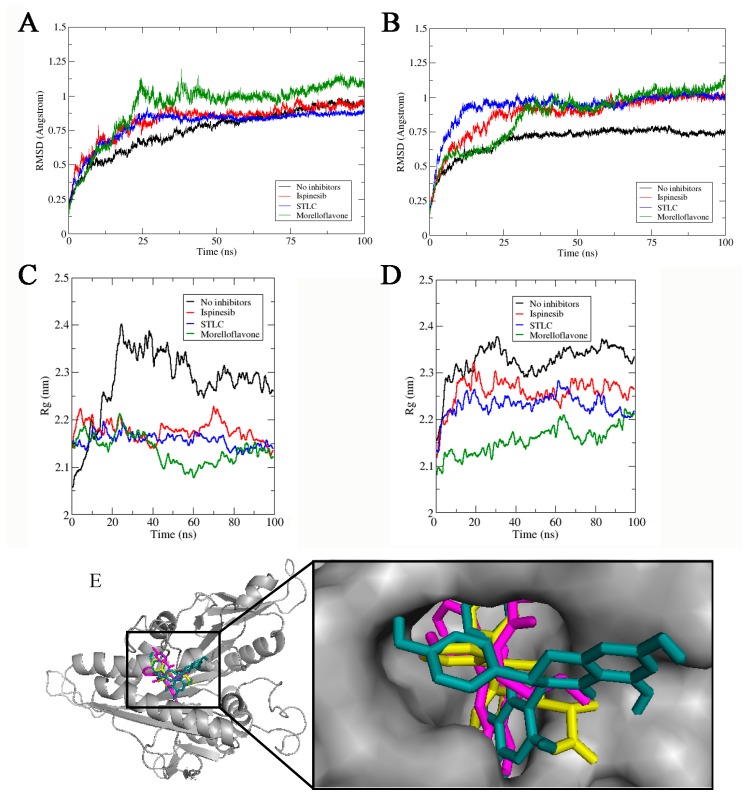
Stability and compactness of Eg5 complexes. RMSD of complexes in the presence of (**A**) ADP, and (**B**) ATP. Rg values of complexes in the presence of (**C**) ADP and (**D**) ATP. (**E**) MF (green stick), STLC (yellow), and ispinesib (magenta) are stably resident within the loop5/α2/α3 binding pocket.

**Figure 4 pharmaceuticals-12-00058-f004:**
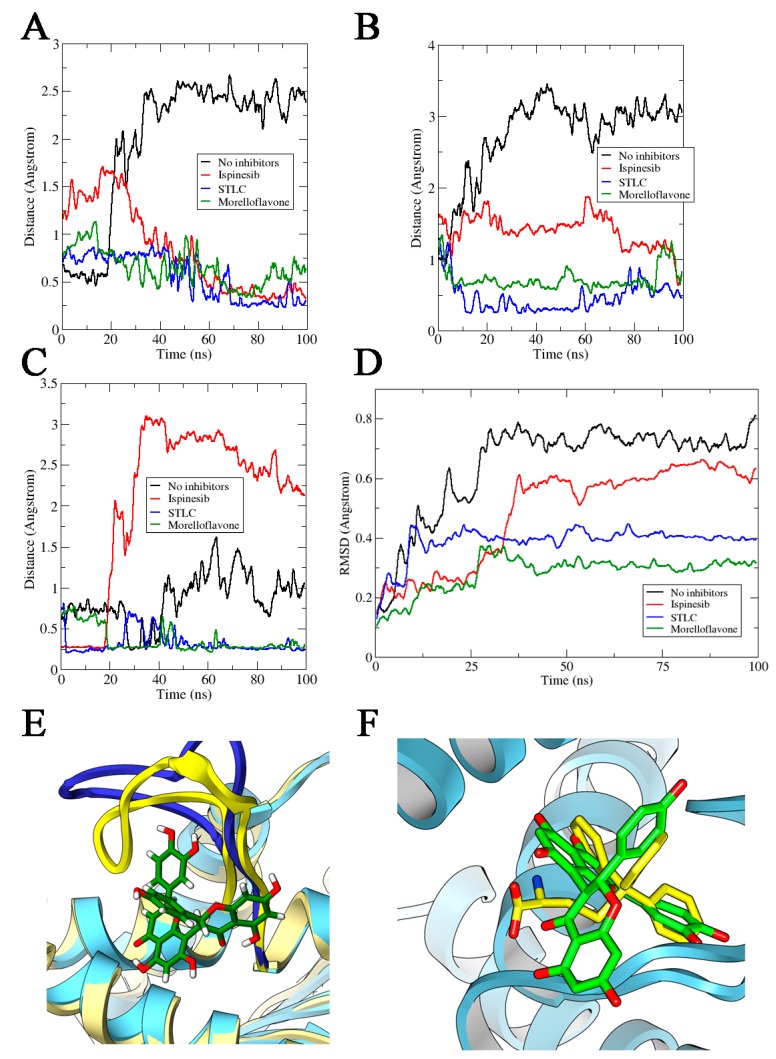
Distances (in Å) between Eg5 Trp127 and Glu215 residues in the presence of (**A**) ADP and (**B**) ATP. (**C**) Estimated distances (in Å) between Trp127 and Tyr211 on Eg5 in the presence of ADP and inhibitors. (**D**) RMSD of loop5 in Eg5-ADP-inhibitor complexes. (**E**) Loop5, yellow cartoon, trapped in the close conformation by inhibitor (MF, reported in green) versus without inhibitor (blue cartoon). (**F**) Comparison between final conformations of STLC (in yellow sticks) and MF (in green sticks) in complex with Eg5 (in blue ribbons) and ADP. The movement of the two inhibitors toward loop5 is comparable.

**Figure 5 pharmaceuticals-12-00058-f005:**
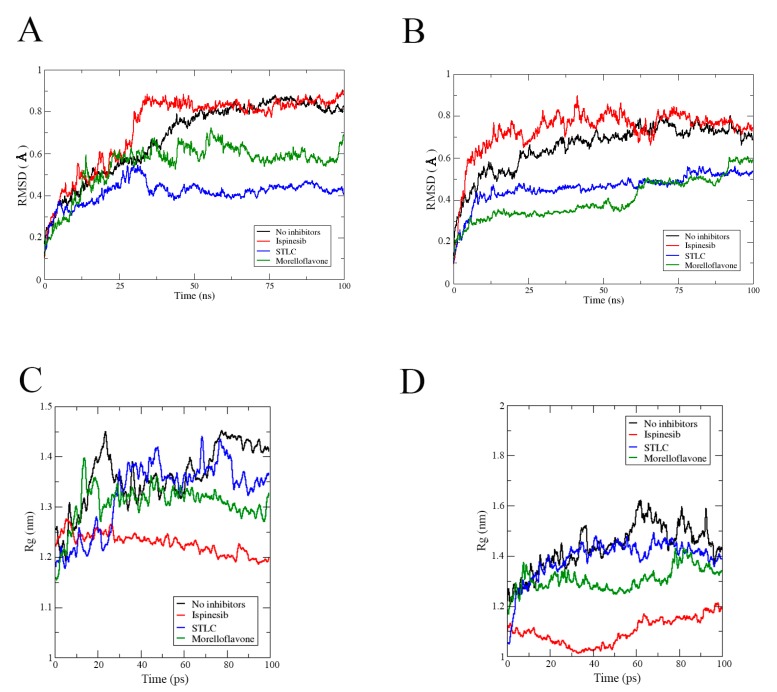
RMSD profiles of the allosteric pocket in Eg5 in the presence of (**A**) ADP and (**B**) ATP. Rg values of the Eg5 allosteric pocket in the presence of (**C**) ADP and (**D**) ATP.

**Figure 6 pharmaceuticals-12-00058-f006:**
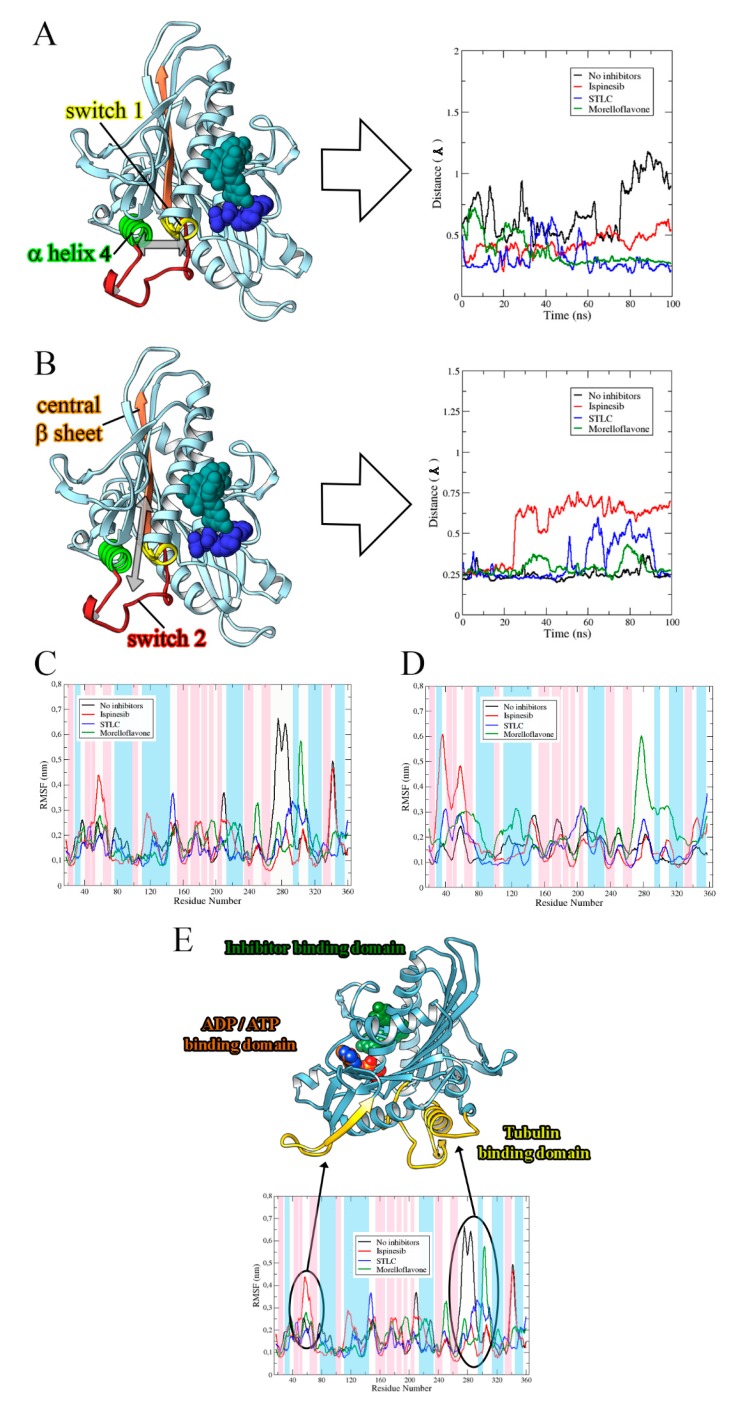
Structural behavior of Eg5 in the presence and absence of inhibitors. Distance of (**A**) switch I from α2-helix and (**B**) switch II from the central β-sheet. Graphs are shown as Eg5 without inhibitors (black), with ispinesib (red), with STLC (blue), and with MF (green). Inhibitor (sphere, cyan) is bound to the allosteric pocket whereas the nucleoside-phosphate (sphere, blue) complexed with Eg5 at the ATP/ADP-binding site. The estimated distance between selected Eg5 segments plotted for STLC and MF are similar. RMSF profiles of (**C**) ADP-bound Eg5 complexes and (**D**) ATP-bound Eg5 complexes. (**E**) Fluctuations in Eg5 regions induced by MF. The β-sheets, α-helices, and loops of secondary structure are highlighted in blue, pink, and white, respectively.

**Figure 7 pharmaceuticals-12-00058-f007:**
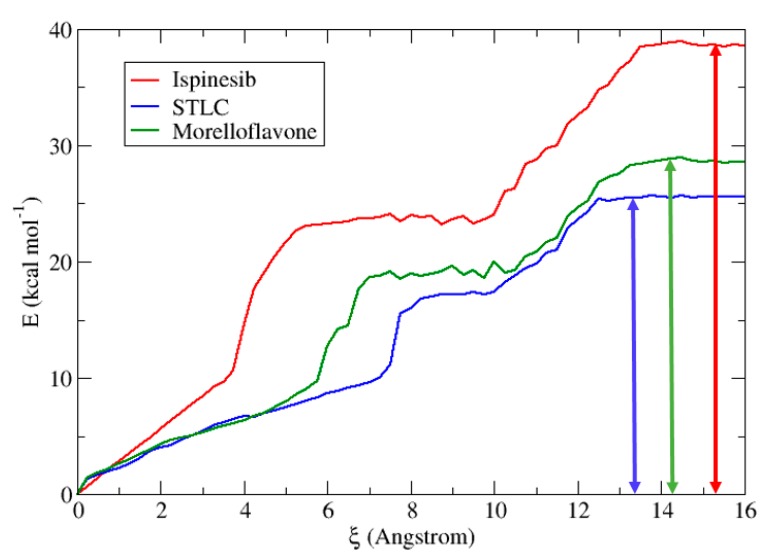
Potential of mean force along the reaction coordinate for the dissociation of inhibitors in Eg5 protein. We show the behavior of ispinesib (red), STLC (blue), and MF (green). The vertical solid double-headed arrows indicate the free energy barriers for the inhibitor dissociations.

**Figure 8 pharmaceuticals-12-00058-f008:**
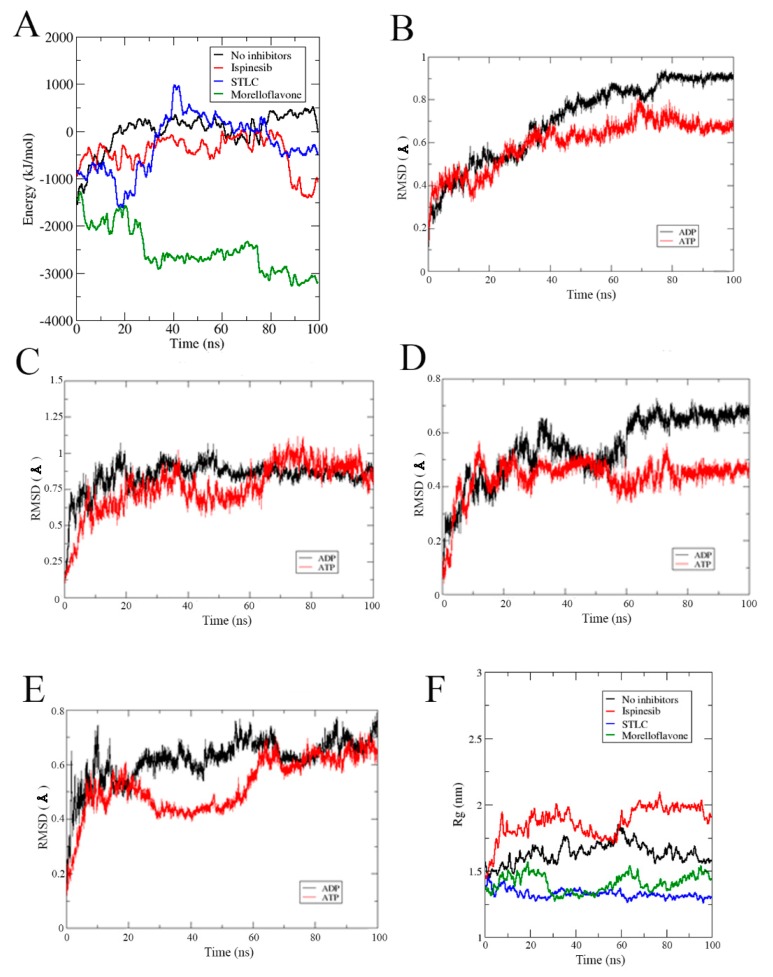
(**A**) Binding affinity of ADP with Eg5 during MD trajectories calculated using the MM/PBSA method on the 100 ns trajectories. RMSD profiles of the active site in (**B**) Eg5-ADP versus Eg5-ATP, (**C**) Eg5-ADP-ispinesib versus Eg5-ATP-ispinesib, (**D**) Eg5-ADP-STLC versus Eg5-ATP-STLC, and (**E**) Eg5-ADP-MF versus Eg5-ATP-MF. (**F**) Rg of Eg5-ATP complexes.

**Table 1 pharmaceuticals-12-00058-t001:** Energy estimation.

Energy (kJ/mol)	Eg5-Ligand Complexes		
	EG5+ADP+Ispinesib	EG5+ADP+STLC	EG5+ADP+MF
Van der Waals	−4.5 ± 8.50	−20.3 ± 9.0	−20.7 ± 8.7
Electrostatic	−103.6 ± 11.0	−158.7 ± 10.7	−156.1 ± 11.1
Polar solvation	−65.1 ± 9.1	−36.1 ± 7.5	−40.1 ± 7.7
Non-polar solvation	126.9 ± 34.2	126.1 ± 34.3	123.2 ± 33.4
Binding energy	−96.31 ± 9.5	−89.0 ± 8.9	−93.7 ± 9.2

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
