# Peer review of "Insights into the Molecular Mechanisms of Eg5 Inhibition by (+)-Morelloflavone"

_pharmaceuticals, 2019, doi:10.3390/ph12020058_

Round 1
Reviewer 1 Report
Ogunwa et al. have submitted an interesting computational study on insights into the molecular mechanisms of Eg5 inhibition by (+)-morelloflavone. Prior to the publication some major revisions should be done as noted below.
1. The Introduction doesn't cover other computational studies with the same protein. Please extend and compare to your study.
2. How were the ligands initially placed into the binding pocket? Docking? Randomly?
3. Figure 2A-B; RMSD values seem too high. Are they in Angstrom and not nm? A simulation with an RMSD around 1 Angstrom (0.1 nm) is stable, but if it is 10 Angstrom (1 nm), there is something wrong.
4. Figure 3A-D; same as previous comment. Wrong unit?
5. Figure 4A-B; same as previous comment.
6. Figure 5A-B; same as previous comment; check the unit.
7. Figure 5C-E; compare the RMSF values to the experimentally determined b-factors. Should be available in the PDB file, or a PDB file can be used on an online server to compute them; e.g., as in Žuvela et al. J. Enzyme Inhib. Med. Chem. 2018, 33, 1430-1443 (doi:10.1080/14756366.2018.1511551). Also label the secondary structure (e.g., beta sheet, alpha helix) on the RMSF plots.
8. Figure 6B-E; re-check the unit.
9. MM/PBSA method is notoriously inaccurate, which is evident from the pretty high standard deviations of your free energy values. Please use a more accurate and precise method such as umbrella sampling. Your simulations (~100 ns) are well equilibrated and you can quickly generate the necessary data. Check: Lemkul and Bevan. J. Phys. Chem. B 2010, 114, 1652-1660 (doi:10.1021/jp9110794) for more information.
Author Response
AUTHORS’ RESPONSES TO REVIEWERS COMMENTS
Dear Editor,
We have revised our manuscript as advised and hereby forward the revised version.
(1) English language proofreading was done on the manuscript before the initial submission. We have also improved the revised version of the manuscript (Areas in red font).
(2) We attached our point-by-point response to each of the reviewers’ comments.
(3) All modified parts are highlighted using red font or purple fonts.
Our appreciation to the reviewers for their valuable suggestions.
Ogunwa et al. have submitted an interesting computational study on insights into the molecular mechanisms of Eg5 inhibition by (+)-morelloflavone. Prior to the publication some major revisions should be done as noted below.
Comment 1: The Introduction doesn't cover other computational studies with the same protein. Please extend and compare to your study.
Response: The introduction section has been revised according to your suggestion, and few sentences on computational studies on Eg5 have been added. Please, see page 2, lines 51-52, 57-63.
Comment 2: How were the ligands initially placed into the binding pocket? Docking? Randomly?
Response: Ligands were docked into the binding pocket (blind docking), as detailed in ‘starting structures’ of the “computational methods” section. Please, see page 15.
Comment 3: Figure 2AB; RMSD values seem too high. Are they in Angstrom and not nm? A simulation with an RMSD around 1 Angstrom (0.1 nm) is stable, but if it is 10 Angstrom (1 nm), there is something wrong.
Comment 4: Figure 3AD; same as previous comment. Wrong unit?
Comment 5: Figure 4AB; same as previous comment.
Comment 6: Figure 5AB; same as previous comment; check the unit.
Response to comment 3 - 6. Thanks for the correction; of course you are right. The units are in Angstrom and we have corrected the Figures (in the new numeration, they are Figure 3-6). Please, see page 5, 8, 12.
Figure 5CE; compare the RMSF values to the experimentally determined bfactors. Should be available in the PDB file, or a PDB file can be used on an online server to compute them; e.g., as in Žuvela et al. J. Enzyme Inhib. Med. Chem. 2018, 33, 14301443 (doi:10.1080/14756366.2018.1511551). Also label the secondary structure (e.g., beta sheet, alpha helix) on the RMSF plots.
Response: The b-factor plot, as estimated on ResQ server, is presented as Figure S3 (see Supplementary Information), and it is compatible with the RMSF. As suggested, appropriate label has been provided for Eg5 secondary structure, as in Žuvela et al. J. Enzyme Inhib. Med. Chem. 2018, 33, 14301443 (doi:10.1080/14756366.2018.1511551). Please, see page 10.
Comment 8: Figure 6BE; recheck the unit.
Response: The units have been corrected in Figure 8 (ex-6). Please, see page 14.
Comment 9: MM/PBSA method is notoriously inaccurate, which is evident from the
pretty high standard deviations of your free energy values. Please use a more accurate and precise method such as umbrella sampling. Your simulations (~100 ns) are well equilibrated and you can quickly generate the necessary data. Check: Lemkul and Bevan. J. Phys. Chem. B 2010, 114,16521660 (doi:10.1021/jp9110794) for more information.
Response: Thanks for your suggestion. The Umbrella sampling method was used to re-estimate the binding energy. We added the new figures and discussion in the manuscript (Page 12). We also modified the material and methods section (section 3.4 page 16).

Reviewer 2 Report
The title seems incomplete; authors may include more information to clearly sell the message, (suggestion, “Morelloflavone inhibits Kinesin spindle protein (Eg5) by….?? what mechanism or process). Ideal title should have message highlight.
Authors explored the potential mechanism of the inhibition with simulation and docking studies. This is exciting and could be important in moving forward with the area of research. The message and interpretations lack in vivo proof and reflects over interpretations of the data. The specific comments are as follow
Major comments
1) There is a concern of novelty, since the same group recently reported the effect of Morelloflavone as a novel inhibitor for mitotic kinesin Eg5 (Ogunwa, T.H. et al., 2018). Novelty in mechanism of action or applied procedure needs to highlight.
2) Figure 2. Eg5-ATP showed a very stable complex with the lowest RMSD compared with Eg5-ADP. However the Rg values of complexes in the presence of ATP or ADP, is not matching with the RMSD values of ATP and ADP, need to explain and justify the data.
3) Figure 4, ispinesib working as a positive control based on the pattern of the RMSD and Rg data compare to other conditions. How authors justify and explain the conformational changes and stability of the morelloflavone with ATP in respect to other conditions.
4) Conclusion (line#423) states that “Future crystallization of the Eg5-ADP-MF complex may verify the described interactions and, elucidate more on the precise molecular structure of the complex”. Figure 2, data shows the stability of Eg5 with ATP complex with lowest RMSD, these needs to consider in the conclusion and future directions. I am gain expecting for in vivo or ex vivo confirmatory data to prove the inhibitory mechanism of Eg5 by morelloflavone.
5) An illustration of study design and outcome would be helpful for readers to easy get the sense of study and message.
Minor comments
1) Result and discussion (line 71) “Among these, the first two biosystems served as controls,” specify the nature of control, ie -/+ control?
2) Figure 1D, there is type error in the figure inset “Morelloflavone”
3) The graphs line’s color of each condition should be consistent throughout in each figure panel.
Author Response
AUTHORS’ RESPONSES TO REVIEWERS COMMENTS
Dear Editor,
We have revised our manuscript as advised and hereby forward the revised version.
(1) English language proofreading was done on the manuscript before the initial submission. We have also improved the revised version of the manuscript (Areas in red font).
(2) We attached our point-by-point response to each of the reviewers’ comments.
(3) All modified parts are highlighted using red font or purple fonts.
Our appreciation to the reviewers for their valuable suggestions.
The title seems incomplete; authors may include more information to clearly sell the message, (suggestion, “Morelloflavone inhibits Kinesin spindle protein (Eg5) by….?? what mechanism or process). Ideal title should have message highlight. Authors explored the potential mechanism of the inhibition with simulation and docking studies. This is exciting and could be important in moving forward with the area of research. The message and interpretations lack in vivo proof and reflects over interpretations of the data. The specific comments are as follow
Response: Eg5 inhibition by morelloflavone follows an allosteric type of inhibition. The mechanisms involve modulation of the protein in some unique manners. This informs our choice of the title “insights into the molecular mechanisms of Eg5 inhibition by (+)-morelloflavone” to appropriately capture the entire findings of the study. In addition, we have provided a detailed abstract with the research highlights.
Major comments
1) There is a concern of novelty, since the same group recently
reported the effect of Morelloflavone as a novel inhibitor for mitotic
kinesin Eg5 (Ogunwa, T.H. et al., 2018). Novelty in mechanism of
action or applied procedure needs to highlight.
Response: In our previous work, we identified morelloflavone as Eg5 inhibitor in vitro. However, the molecular mechanism of the inhibition, dynamics of Eg5-morelloflavone interaction, and the enzyme behavior in the presence of the inhibitor were not investigated. To understand these mechanisms, we carried out computational experiments reported in this manuscript. The results showed morelloflavone induced allosteric α2/α3/L5 pocket closure and stability, alteration in the affinity of nucleotides to the active site, conformational modulation of the microtubule binding site, reduced flexibility of Eg5 structure and shifts in various regions of the protein including α2, α3 helices, β6 sheet. Please, see the sentences highlighted in purple font in the abstract (page 1) and introduction (page 2).
2) Figure 2. Eg5ATP showed a very stable complex with the lowest RMSD compared with Eg5ADP. However, the Rg values of complexes in the presence of ATP or ADP, is not matching with the RMSD values of ATP and ADP, need to explain and justify the data.
Response: We have included the following sentences in our manuscript to explain the difference in stability between Eg5-ATP and Eg5-ADP complexes.
“The difference in the nucleotides structure, i.e., the presence of a γ-phosphate in ATP which is missing in ADP, may contribute to the variation in the RMSD values”. Please, see page 4.
The absence of the γ-phosphate in ADP permits the Eg5-ADP structure to adopt a range of diverse conformations. Please, see page 10 (red font).
Regarding the Rg, we have edited the sentences as “….for the corresponding Eg5-ATP-MF complex, the Rg increased from 2.06 nm to 2.21 nm throughout the simulation (Figure 3D), which was lower than that of Eg5-ATP. The presence of γ-phosphate of ATP might play a key role in the Rg increase. For STLC- and ispinesib-bound Eg5-ATP structures, Rg values were also lower compared to Eg5-ATP”.
We wish to mention that we provided detailed explanation on the implication of γ-phosphate of ATP in page 10, such as “The structural implications of the ATP γ-phosphate include a physical contact between Switch-II and the γ-phosphate of ATP during the simulation, possibly enhancing the rigidity of Eg5-ATP structure. These observations may underlie the variation associated with the Rg and RMSD plot pattern”.
3) Figure 4, ispinesib working as a positive control based on the pattern of the RMSD and Rg data compare to other conditions. How authors justify and explain the conformational changes and stability of the morelloflavone with ATP in respect to other conditions.
Response: The stability observed in Eg5-ADP-MF and Eg5-ATP-MF in this study shows that the inhibitor might bind the protein at the allosteric pocket in the presence of the nucleotide to suppress ATP hydrolysis. Morelloflavone is not competitive with nucleotide binding on Eg5. The conformational alterations observed in the presence of the nucleotides contribute the inhibition of the ATPase activity of Eg5.
4) Conclusion (line#423) states that “Future crystallization of the Eg5ADPMF complex may verify the described interactions and, elucidate more on the precise molecular structure of the complex”. Figure 2, data shows the stability of Eg5 with ATP complex with lowest RMSD, these needs to consider in the conclusion and future directions. I am gain expecting for in vivo or ex vivo confirmatory data to prove the inhibitory mechanism of Eg5 by morelloflavone.
Response: We specifically suggest Eg5-ADP-MF complex for crystallization because no Eg5-ATP crystal structure has been deposited in the protein data bank till date. It seems very difficult to crystallize Eg5 in the ATP bound state, maybe due to the rate of ATP hydrolysis to form ADP and Pi. As evidence, all available Eg5-inhibitor crystal structures are co-crystallized with ADP. However, we had modified the last sentences in “Conclusion” as follows:
“In vivo experiments to confirm the inhibitory mechanisms of morelloflavone against Eg5 and crystallization of Eg5-nucleotide-MF shall be undertaken in our subsequent research work to gain more insights on the evolution and biochemical implication of the molecular structure of the complexes”.
Please, see page 16.
5) An illustration of study design and outcome would be helpful for readers to easy get the sense of study and message.
Response: The suggested illustration has been added to the manuscript. Please, see page 3.
Minor comments
1) Result and discussion (line 71) “Among these, the first two biosystems served as controls,” specify the nature of control, ie /+ control?
Response: The sentence has been edited as “Among these, the first two biosystems served as negative controls”. Please, see page 3.
2) Figure 1D, there is type error in the figure inset “Morelloflavone”
Response: The typographical error in Figure 3 has been corrected. Please, see page 5.
3) The graphs line’s color of each condition should be consistent throughout in each figure panel.
Response: The graphs have been revised. The graphs line’s colour are now uniform.
Reviewer 3 Report
1/
page 2 line 73: I would add here also literature references 14 and 15 to the text
2/
Figure 6A
It is not explained in the text how those energies are calculated. Are they extracted from the MD log files?
Author Response
AUTHORS’ RESPONSES TO REVIEWERS COMMENTS
Dear Editor,
We have revised our manuscript as advised and hereby forward the revised version.
(1) English language proofreading was done on the manuscript before the initial submission. We have also improved the revised version of the manuscript (Areas in red font).
(2) We attached our point-by-point response to each of the reviewers’ comments.
(3) All modified parts are highlighted using red font or purple fonts.
Our appreciation to the reviewers for their valuable suggestions.
1. page 2 line 73: I would add here also literature references 14 and 15 to the text
Response: Reference 14 and 15 are now inserted as suggested. Please, see page 2.
2. Figure 6A; It is not explained in the text how those energies are calculated. Are they extracted from the MD log files?
Response: The energy values plotted in Figure 8 (ex 6) are calculated from the MD log files (MM/PBSA method) and using the GROMACS 5 analysis toolkit. We added an explanation on the Figure caption. Please, see page 16.
Round 2
Reviewer 1 Report
The authors have addressed all my comments and concerns. After English proofing, the manuscript can be accepted for publication.
Reviewer 2 Report
Most of the concerns are well taken and justified.